# Differences in Expression of Selected Interleukins in HIV-Infected Subjects Undergoing Antiretroviral Therapy

**DOI:** 10.3390/v14050997

**Published:** 2022-05-07

**Authors:** Beata Szymańska, Karolina Jurkowska, Brygida Knysz, Agnieszka Piwowar

**Affiliations:** 1Department of Toxicology, Faculty of Pharmacy, Wroclaw Medical University, 50-556 Wroclaw, Poland; beata.szymanska@umw.edu.pl (B.S.); agnieszka.piwowar@umw.edu.pl (A.P.); 2Department of Infectious Diseases, Liver Diseases and Acquired Immune Deficiencies, Faculty of Medicine, Wroclaw Medical University, 50-367 Wroclaw, Poland; brygida.knysz@umw.edu.pl

**Keywords:** interleukin-4, interleukin-7, interleukin-15, HIV, combined antiretroviral therapy

## Abstract

The use of combined antiretroviral therapy (cART) inhibits the replication of the Human Immunodeficiency Virus (HIV) and thus may affect the functioning of the immune system, e.g., induce changes in the expression of certain cytokines. The aim was to examine the effect of cART on the expression of selected cytokines: interleukin -4, -7 and -15 in HIV-infected subjects. The test material was the plasma of HIV-infected men and healthy men (C, control group). The levels of interleukin were measured by immunoenzymatic method before cART and one year after treatment in relation to the C group. HIV-infected men were analyzed in subgroups depending on the HIV-RNA viral load, CD4^+^ and CD8^+^T-cell counts, and the type of therapeutic regimen. A significantly higher level of IL-4 was demonstrated in HIV-infected men before cART compared to those after treatment and in the control group. The use of cART resulted in a significant decrease in the level of IL-7 in HIV-infected men; however, high levels of IL-7 were associated with a low number of CD4^+^ T cells and CD8^+^ T cells. An increase in the level of IL-15 in HIV-infected men was noted after the use of cART. There was no difference in the expression of interleukins depending on the treatment regimen used. The study showed the effect of cART on the expression of interleukins, especially IL-4 and IL-7. Further research in this direction seems promising, confirming the role of these interleukins in the course of the disease.

## 1. Introduction

The pathogenesis of HIV infection and the development of AIDS are a consequence of the infectious properties of the virus and the host’s immune response to the virus. The balance between the effectiveness of these two components determines the outcomes of infection, from the development of AIDS to long-term survival [1,2].

The discovery of the multistage HIV replication cycle in human CD4^+^ T cells has identified potential drug targets for arresting or slowing down the process of viral replication. The first approved drug was zidovudine, a nucleoside reverse transcriptase inhibitor (NRTI). Although NRTI monotherapy was shown to reduce viral load, delay disease progression, and prolong survival, the use of a single agent did not result in sustained viral suppression. The use of three HIV protease inhibitors (PI) radically changed the course of the HIV epidemic. The administration of a PI and two NRTI therapies resulted in the rapid reduction of HIV, improvement in immune function, regression of difficult-to-treat opportunistic infections such as Kaposi’s sarcoma and progressive multifocal leukoencephalopathy, and reduced mortality. From that moment forward, combined antiretroviral therapy (cART) became the basis of the treatment of HIV infection. The strategic use of three drugs from each of the two classes enables the achievement and maintenance of an undetectable viral load, effectively bringing the disease into remission. The most common combinations are two NRTIs and one non-nucleoside reverse transcriptase inhibitor (NNRTI) or two NRTIs and one PI [3,4].

The immune response is tightly regulated by various subsets of CD4^+^ Th cells (lymphocytes T CD4^+^ helper). T-cell clones show at least one of two distinct cytokine production profiles, including CD4^+^ Th1 cells and CD4^+^ Th2 cells. The cytokines produced by Th1 stimulate macrophages and activate CD8^+^ cells, thereby promoting a response based on traditional cellular immunity. The cytokines produced by Th2 cells activate eosinophil function and Th cells to produce immunoglobulin G1 and immunoglobulin E antibodies, thereby promoting response to traditional humoral immunity and suppressing cellular immunity [4,5].

While the mechanisms underlying Th1 cell dominance over Th2 are not fully understood, it is clear that Th1 cells are prevalent in immunologically healthy individuals. During HIV infection and the resulting disturbance of immune regulation, the Th2 cells’ response is superior to the Th1 cells’ response. Thus, cellular immunity is greatly reduced and ultimately destroyed during HIV-induced immunosuppression [6].

The use of cART has an inhibitory effect on the replication of the virus and thus may affect changes in the functioning of the immune system, e.g., changes in the expression of certain cytokines (interleukins) in HIV-infected subjects [7].

Interleukin 4 (IL-4) is a pleiotropic cytokine produced by Th2 cells, mast cells and basophils. Its activity is broad, affecting many populations of cells of the immune system and showing antagonistic effects (in most cases) to interferon-γ. It also affects T cells, directing their development towards Th2 cells. IL-4 may inhibit cellular and humoral immunity [8]. IL-4 regulates the differentiation of precursor T helper cells into those of the Th2 subset that mediate humoral immunity and modulate antibody production. With respect to HIV-1 infection, IL-4 was reported to differentially regulate two major HIV-1 coreceptors, CXCR4 for SI (syncytium-inducing) variants and CCR5 for NSI (no syncytium-inducing) viruses. IL-4 down-regulates CCR5 expression and thus inhibits replication of HIV-1 NSI isolates in human T-cells and macrophages.

On the other hand, IL-4 upregulates the expression of CXCR4. In addition, IL-4 stimulates the expression of HIV-1 through the activation of viral transcription. The combination of these effects of IL-4 on HIV-1 replication may be involved in the phenotypic switch from NSI to SI as well as disease progression in HIV-1 infection. Thus, IL-4 could be an important factor in viral evolution and AIDS pathogenesis [9].

Interleukin 7 (IL-7) is a hematopoietic growth factor secreted by stromal cells in the bone marrow and thymus. It is also produced by keratinocytes, dendritic cells, hepatocytes, neurons, and epithelial cells but is not produced by lymphocytes. IL-7 is needed for the development and maintenance of T cells and the restoration of mature T-cell homeostasis. Under normal conditions, its resource is relatively limited [10,11]. IL-7 plays an important role in immune system function, especially in the development of T cells, including CD4^+^ cells. IL-7 may improve HIV-specific immune responses by increasing the number of CD4^+^ cells and boosting immune response. Recently, IL-7 therapy has been demonstrated to favorably impact T-cell functions by promoting their proliferation and expansion in subjects receiving cART. However, the effects of IL-7 administration on the mechanisms of HIV persistence and the size of the latent reservoir are still unclear [12,13].

Interleukin 15 (IL-15) is a pleiotropic and multifunctional cytokine that has a diverse range of biological effects in the body produced by monocytes, macrophages, and dendritic cells. It plays a key role in the host’s defense against viral and non-viral intracellular pathogens. IL-15 induces the proliferation of natural killer cells and their antibody-dependent cytotoxicity and stimulates their production of cytokines and chemokines. It activates neutrophils, acts on activated T cells, and is involved in the maintenance of memory T cells. It is involved in the differentiation of dendritic cells and induces angiogenesis and muscle differentiation [14,15]. IL-15 plays an essential role in the development and maturation of NK. This interleukin enhances the cytolytic potential of NK cells. NK cells play an important role in host defense by directly killing virus-infected cells and activating macrophages through INF-γ synthesis. Thus, the production of an optimum level of IL-15 may represent a prerequisite for normal maintenance of NK activity, the antimicrobial potential of macrophages, and the induction of pathogen-specific immune responses. Virus-induced IL-15 gene activation is linked to the massive proliferation of CD8^+^ T cells and the maintenance of virus-specific memory T cells (15). IL-15 has also been shown to substitute for CD4^+^ T-cell help in the induction and maintenance of antiviral CTL (cytotoxic T lymphocytes) responses (16). This IL-15 function may be crucial in HIV infection, particularly in later stages, when CD4^+^ T-cell numbers decline precipitously [14,16].

In the authors’ research, an attempt was made to assess the expression of three selected interleukins, IL-4, -7 and -15, in HIV-infected men and to examine the effect of one year of antiretroviral therapy on their change. The analyses took into account the influence of parameters characterizing the state of the immune system, such as HIV-RNA viral load, CD4^+^ and CD8^+^ T-cell counts, as well as the type of antiretroviral treatment regimen used on the blood expression of the IL-4, IL-7 and IL-15. The interdependencies between interleukins were also analyzed to investigate the contribution of these cytokines to HIV infection.

## 2. Individuals and Methods

### 2.1. Participants’ Selection and Collection of Data

The research material was plasma obtained from whole blood samples taken from 30 HIV-infected subjects (men) treated at the Center for Prevention and Treatment of Infectious Diseases and Addictions and the Department of Infectious Diseases, Liver and Acquired Immunity Defects of the Wroclaw Medical University.

In the group of HIV-infected men, inclusion criteria were the patient’s consent for tests and confirmation of the presence of HIV infection. Exclusion criteria were diseases such as diabetes, cancer, hypertension, hepatitis B or C virus infection, especially urinary tract diseases, and concomitant use of drugs other than cART. The control group consisted of 28 HIV-negative healthy volunteers (men). 

HIV-infected men were treated with two therapeutic regimens, which included two NRTIs (emtricitabine and tenofovir alafenamide) in combination with PIs (ritonavir-boosted lopinavir or cobicistat-boosted darunavir) or INSTIs (dolutegravir). 

Patients were also analyzed in subgroups depending on HIV-RNA viral load (below and above 100,000 RNA copies/mL), CD4^+^ T-cell count (below and above 300 cells/µL), CD8^+^ T-cell count (below and above 1000 cells/µL), and the type of therapeutic regimen (INSTIs or PIs) both before and after one year of anti-HIV therapy.

Data on HIV-RNA viral load levels, CD4^+^ and CD8^+^ T-cell counts were obtained from medical records. HIV-RNA isolation was performed using a system viral nucleic acid kit (Roche Diagnostics, Mannheim, Germany). HIV-1 viral load was measured by real-time PCR assay (COBAS TaqMan HIV-1 test v. 2.0, Roche Diagnostics, Basel, Switzerland). The detection limit was 40 copies/mL. The CD4^+^ and CD8^+^ T-cell counts were tested by flow cytometry using the FACSCount Becton Dickinson system (BD Biosciences, San Jose, CA, USA).

### 2.2. Blood Collection and Enzyme-Linked Immunosorbent Assay (ELISA)

Blood was drawn from HIV-infected men twice—the first time prior to the initiation of antiretroviral therapy and again one year after the initiation of treatment.

Fasting blood samples were collected from HIV-infected men and from healthy (HIV-uninfected) men in anticoagulant tubes (5 mL blood, containing 1.6 mg/mL EDTA, Sarstedt, Poland). The samples were centrifuged at 1500× *g* for 10 min to separate plasma. Plasma was aliquoted and frozen at −80 °C until assayed.

The concentration of the studied interleukins was measured using the enzyme-linked immunosorbent assay method (ELISA). The following enzyme immunoassays were used in the studies: Human IL-4 Immunoassay (R&D Systems, Inc., Minneapolis, MN, USA); Human IL-7 ELISA Kit (Thermo Fisher Scientific Inc., Waltham, MA, USA); Human IL-15 Immunoassay (R&D Systems, Inc., Minneapolis, MN, USA), according to the manufacturer’s instructions. Absorbance was read at 450 nm with a microplate reader (STAT FAX 2100, Awareness Technology, Inc., Palm City, FL, USA).

All participants were informed about the purpose of the study and gave their written consent. The study was approved by the Ethics Committee of the Medical University in Wroclaw (KB-597/2019) and with the Helsinki Declaration. 

### 2.3. Statistical Analysis

The statistical analysis of the obtained results was performed using Statistica 13.3 PL by StatSoft. The Kolmogorov–Smirnov and Lilliefors test was used to check the data distribution. Data distributions of tested interleukins (IL-4, IL-7, IL-15) were non-parametric; therefore, Mann–Whitney *U*-tests were selected for the analysis. Correlations between parameters were assessed using the Spearman test. The comparison of results within subgroups was performed using the Wilcoxon test. In all analyzes, *p* < 0.05 was considered a significant value.

## 3. Results

### 3.1. Study Population

The number of HIV-infected men (*n* = 30) was similar to that of men not infected with HIV (*n* = 28). The mean age was 33 years (25–54) among HIV—infected men and 36 (22–58) in the control group (C, HIV(-) men), and did not differ statistically (*p* > 0.05).

### 3.2. Assessment of Immunological Parameters in HIV-Infected Men

The assessment parameters such as HIV viral load, CD4^+^ and CD8^+^ T-cell counts, characterizing HIV- infected men before cART (A) treatment and after one year of cART (B) are presented in Table 1.

The number of HIV-RNA copies and the amount of CD4^+^ and CD8^+^ T cells were statistically significantly different between the groups of patients before cART (A) compared to the values of these parameters obtained in the group of patients after therapy (B) (Table 1).

In HIV-infected men before cART, the mean HIV-RNA viral load was more than 15,000-fold higher than the values obtained after one year of cART. 

The mean amount of CD4^+^ T cells after cART increased 1.7-fold relative to CD4^+^ T-cell count prior to administration of antiretroviral therapy.

Treatment with cART decreased (2.2-fold) the mean CD8^+^ T-cell count compared to untreated HIV-infected men.

The CD4^+^/CD8^+^ ratio in HIV-infected men after cART was 2.2-fold higher compared to the CD4^+^/CD8^+^ ratio in HIV-infected men before cART. 

### 3.3. Interleukins (IL-4, -7, -15) in Plasma of HIV-Infected Men before cART Therapy, after cART Therapy and in the Control Group

The obtained values of interleukins (IL-4, IL-7, and IL-15) in HIV-infected men before cART, after cART, and in the control group are presented in Table 2.

The mean plasma IL-4 concentration value of HIV-infected men before (A) cART was twice as high as in healthy uninfected control men (C). There was a statistically significant difference between IL-4 concentrations in these patients relative to the control (*p* = 0.03). The mean concentration of IL-4 in the group of HIV-infected men after cART (B) therapy was 1.3-fold higher than the value obtained in the control group, but the difference was not statistically significant. A statistically significant difference was found between the values of IL-4 concentrations in HIV-infected men before cART (A) compared to HIV-infected men after antiretroviral treatment (B), (*p* = 0.04).

The mean value of IL-7 plasma concentration in HIV-infected men before (A) cART was almost 1.5-fold higher compared to cART (B). A statistically significant difference was demonstrated between the values of IL-7 concentrations in the HIV-infected before cART (A) compared to men after cART (B). A statistically significant difference was demonstrated (*p* = 0.02). The mean concentration of IL-7 in the group of HIV-infected men before cART (A) was also 1.5-fold higher compared to the value obtained in the control group, but the difference was not statistically significant. Mean values of IL-7 concentration among men after cART (B) and in the control group were at a similar level, and no statistically significant difference was found.

The mean value of IL-15 concentration in the plasma of HIV-infected men before treatment (A) was similar to the mean value of IL-15 concentration obtained in the control group. There was no statistically significant difference between these groups. The mean value of IL-15 concentration in men after treatment (B) in comparison to the mean value of IL-15 concentration in the control group was 1.6-fold higher. There was no statistically significant difference between IL-15 levels in men after cART (B) compared to men in the control group. The pre-treatment IL-15 concentration value in HIV-infected men (A), compared to HIV-infected men after treatment (B), was 1.4 times lower, but no statistically significant difference was found.

### 3.4. Results of IL-4, -7 and -15 before and after cART in HIV-Infected Men by HIV-RNA Viral Load

The analysis of IL-4, IL-7 and IL-15 was performed in subgroups of HIV-infected men both before cART (A) and after treatment (B), depending on the amount of HIV-RNA copies (below and above 100,000 copies/mL). The results are presented in Table 3.

Analyzing the values of IL-4 levels in HIV-infected men before cART (A) depending on the number of HIV-RNA copies ≤ 100,000/mL and >100,000/mL, a statistically significant difference was demonstrated (*p* = 0.001). The median IL-4 was 1.8-fold higher in the subgroup of HIV -infected men with a HIV-RNA > 100,000/mL copy count.

In HIV-infected men with HIV-RNA ≤ 100,000 copies/mL] (A) the median levels of IL-7 and IL-15 were 1.2-fold and 1.6-fold higher compared to the median levels of these interleukins in HIV-infected men with HIV-RNA > 100,000/mL, respectively, but no statistically significant differences were found.

There was no difference in IL-4, -7 and -15 levels in HIV-infected men before cART (A) with HIV-RNA ≤ 100,000 [copies/mL] compared to these interleukins in HIV-infected men with HIV-RNA ≤ 100,000 copies/mL after cART (B).

HIV-infected men with a viral load of HIV-RNA > 100,000 [copies/mL] were not found to be present after cART (B).

Additionally, interleukin levels were compared in HIV-infected men after cART (B) with no HIV detected (N = 19) to HIV-infected men with low HIV levels detected (N = 11; a viral load of HIV-RNA mean 40 copies/mL).

The medians (IQR) for IL-4, -7, and -15 in plasma of HIV-infected men after cART (B) who had a mean HIV-RNA value of 40 copies/mL were 1.80 (0.50–3.70), 260.20 (61.40–720.60) and 1.90 (1.40–3.40), respectively. The medians (IQR) for IL-4, -7 and -15 in the plasma of HIV-infected men after cART (B) with no HIV-RNA detected were 3.50 (1.80–5.90), 128.10 (49.20–770.00) and 2.7 (1.80–3.70), respectively. The differences between interleukin levels in HIV-infected men with a low viral load of HIV-RNA and interleukin levels in men with undetectable HIV were not statistically significant.

### 3.5. Results of IL-4, -7 and -15 before and after cART in HIV-Infected Men by CD4^+^ T Cell Count

The analysis of IL-4, IL-7 and IL-15 was performed in subgroups of HIV-infected men both before cART (A) and after treatment (B), depending on the CD4^+^ T-cell count (below and above 300 cells/µL). The results are presented in Table 4.

In HIV-infected men with CD4^+^ T cell ≤ 300 count [cells/µL] before cART (A), the median levels of IL-4 and IL-15 were similar to the median levels of these interleukins in HIV-infected men with a CD4^+^ T cell > 300 count [cells/µL] (A) and were not statistically significantly different. 

The median level of IL-7 in HIV-infected men with a CD4^+^ T cell ≤ 300 count [cells/µL] (A) was 2.3-fold higher compared to HIV -infected men with a CD4^+^ T cell > 300 count [cells/µL] (A), and the difference was shown to be statistically significant (*p* = 0.026).

In HIV-infected men with a CD4^+^ T cell ≤ 300 count [cells/µL] after cART (B), the median levels of IL-4 and IL-15 were 1.9-fold and 1.4-fold higher. IL-7 was 0.7-fold lower compared to the median levels of these interleukins in HIV-infected men with a CD4^+^ T cell > 300 count [cells/µL] (B), but no statistically significant differences were found.

There was no difference in IL-4, IL-7 and IL-15 levels in HIV-infected men before cART with CD4^+^ T cell ≤ 300 count [cells/µL] (A) compared to these interleukins in HIV-infected men with CD4^+^ T cell ≤ 300 count [cells/µL] after cART (B). Similarly, no difference was demonstrated in IL-4, -7 and -15 levels in HIV-infected men before cART with CD4^+^ T cell > 300 count [cells/µL] (A) compared to these interleukins in HIV- infected men with CD4^+^ T cell > 300 count [cells/µL] after cART (B). 

### 3.6. Results of IL-4, -7 and -15 before and after cART in HIV-Infected Men by CD8^+^ T Cell Count

The analysis of IL-4, IL-7 and IL-15 was performed in subgroups of HIV-infected men both before cART (A), and after treatment (B), depending on the CD8^+^ T-cell count (below and above 1000 cells/µL). The results are presented in Table 5.

In HIV-infected men with a lymphocyte T CD8^+^ count ≤ 1000 [cells/µL] (A), the median levels of IL-4 and IL-15 were 1.1-fold and 2.3-fold higher, respectively, than the median levels of these interleukins in men with a CD8^+^ T-cell count > 1000 [cells/µL] (A) and were not statistically significantly different. 

The median level of IL-7 in HIV-infected men with a CD8^+^ T-cell count ≤ 1000 [cells/µL] (A) was 6-fold higher compared to HIV-infected men with a CD8^+^T-cell count > 1000 [cells/µL] (A), and the difference was shown to be statistically significant (*p* = 0.027).

In HIV-infected men with a CD8^+^ T-cell count ≤ 1000 [cells/µL] after cART (B), the median levels of IL-4 and IL-15 were higher by 1.8-fold and 1.4-fold, but IL-7 was 1.5-fold lower compared to the median levels of these interleukins in HIV-infected men with CD8^+^ T-cell count > 1000 [cells/µL] (B) respectively, but no statistically significant differences were found.

There was no difference in IL-4, IL-7 and IL-15 levels in HIV-infected men before cART with a CD8^+^ T-cell count ≤ 1000 [cells/µL] (A) compared to these interleukins in men with a CD8^+^ T-cell count ≤ 1000 [cells/µL] after cART (B). Similarly, no difference was demonstrated in IL-4, IL-7 and IL-15 levels in HIV-infected men before cART with CD8^+^ T-cell count > 1000 [cells/µL] (A) compared to these interleukins in men with a CD8^+^ T-cell count > 1000 [cells/µL] after cART (B).

### 3.7. Results of IL-4, -7 and -15 before and after cART in HIV-Infected Men in Terms of the CD4^+^/CD8^+^ Ratio

In group A, before cART, the CD4^+^/CD8^+^ ratio ≤ 1.00 had all HIV-infected men. In group B, after cART, the CD4^+^/CD8^+^ ratio > 1.00 had 6 (20%) HIV- infected men. The analysis was performed only in group B after cART.

Median (IQR) for IL-4, -7 and -15 in plasma of HIV-infected men after cART (B) in the subgroup with CD4^+^/CD8^+^ ratio ≤ 1.00 were 3.45 (1.7–4.9) pg/mL, 163.60 (66.35–751.35) pg/mL and 2.65 (1.85–3.75) pg/mL, respectively.

Median (IQR) for IL-4, -7 and -15 in plasma of HIV-infected men after cART (B) in the subgroup with CD4^+^/CD8^+^ ratio > 1.00 were 1.60 (1.10–1.90) pg/mL, 162.40 (61.40–339.7) pg/mL and 1.75 (1.60–2.90) pg/mL, respectively.

The level of IL-4 in HIV-infected men (B) having a CD4^+^/CD8 ^+^ ratio ≤ 1, was higher at 2.2-fold compared to HIV-infected men with CD4^+^/CD8^+^ > 1 and was statistically significant (*p* = 0.04).

The level of IL-7 in HIV-infected men (B) having a CD4^+^/CD8^+^ ratio ≤ 1 was similar to that of HIV-infected men with CD4^+^/CD8^+^ > 1.

The level of IL-15 in HIV-infected men (B) having a CD4^+^/CD8^+^ ratio ≤ 1, was higher at 1.5-fold compared to HIV-infected men with CD4^+^/CD8^+^ > 1, but not was statistically significant.

### 3.8. IL-4, -7 and -15 Results in HIV-Infected Men Subjected to cART with Protease Inhibitors (PIs) and Integrase Transfer Inhibitors (INSTIs) Treatment

Medians and interquartile ranges for IL-4, IL-7 and IL-15 in the plasma of HIV- infected men after cART subgroups with Protease inhibitors (PIs) and Integrase transfer inhibitors (INSTIs) treatment are presented in Table 6. 

The antiretroviral regimen used, whether cART with Protease inhibitors or Integrase transfer inhibitors treatment, had no significant effect on IL-4, IL-7 and IL-15 levels.

### 3.9. Correlations between Interleukins

The analysis of mutual linear relationships between the studied interleukins showed the presence of three significant correlations. These correlations are presented in Figure 1A,B.

A negative correlation was found between the number of CD4^+^ T cells in the group of HIV-infected men before cART (A) and the concentration of IL-7 in the plasma of these HIV-infected men. The Spearman R coefficient was R = −0.408, (*p* = 0.027). 

In the subgroup of HIV-infected men after cART (B) therapy, a mutually positive relationship between IL-4 and IL-15 was demonstrated. The value of the Spearman R coefficient = 0.590 (*p* = 0.0002).

Visualizations of the IL-4, -7 and -15 results obtained are shown in Appendix A.

## 4. Discussion

In the authors’ research, an attempt was made to evaluate the expression of three selected interleukins important to the course of the disease—IL-4, -7 and -15—in HIV-infected men to investigate the effect of the applied therapy in a one-year follow-up. 

IL-4 synthesis is stimulated during the development of AIDS as a result of HIV-induced domination of Th2 cell stimulation [17]. This thesis is reflected in the results obtained in our study, where a significantly higher level of IL-4 was observed in untreated HIV-infected patients compared to patients treated with cART (*p* = 0.04) and in the control group (*p* = 0.03). The concentration of IL-4 in HIV-infected men after one year of cART was comparable to the concentration of IL-4 in uninfected men.

Osuji et al. [7] investigated the levels of pro-inflammatory and anti-inflammatory cytokines in serum in people infected with HIV. The study results showed significant differences in the levels of anti-inflammatory cytokines IL-4, IL-10 and TGF-β before starting cART therapy, 6 months after cART and 1 year after cART treatment. Anti-inflammatory cytokine levels, including IL-4, were increased in pre-treatment HIV-infected patients compared with 6 and 12 months of treatment and compared to the control group. IL-4 and IL-10 showed no significant difference after 1 year of cART treatment compared to the control group.

IL-4 promotes a number of immune functions that influence macrophage differentiation, the differentiation of CD4^+^ T cells into Th2 cells, and the inhibition of the secretion of various pro-inflammatory cytokines [18]. The results suggest a link between an increase in IL-4 production during retrovirus-induced immunosuppression and the suppression of cellular immune responses [19].

Our own studies showed a significantly higher level of IL-7 in HIV-infected men before cART compared to the level of this interleukin in patients after therapy and in the control group. It was noted that a higher level of IL-7 was associated with a lower count of CD4^+^ and CD8^+^ T cells in HIV-infected patients prior to cART, which was further confirmed by a negative correlation between IL-7 and CD4^+^ T cells.

IL-7 is implicated in thymopoiesis and regulates peripheral naive T-cell survival by modulating the expression of the anti-apoptotic molecule Bcl-2 and sustains peripheral T-cell expansion in response to antigenic stimulation. Infection with HIV leads to T lymphopenia and immune dysfunction. Increased IL-7 plasma levels are observed in HIV-infected patients. The existence of an inverse correlation between IL-7 plasma levels and the CD4^+^ T-cell count suggests that a direct feedback mechanism is working to restore peripheral T-cell counts in lymphopenic patients. Here, IL-7 plasma levels are a good predictive marker of CD4^+^ T-cell restoration during therapy. The cART considerably slows disease progression, decreases the viral load, and significantly increases peripheral CD4^+^ T-cell counts [20].

Progressive HIV infection is associated with a complex deregulation of the IL-7/IL-7R receptor pathways, including increased plasma levels of IL-7 with a decrease in the percentage of CD4^+^ and CD8^+^ T cells expressed by the CD127 receptor [21].

Hodge et al. [22] explored how immune reconstitution through antiretroviral therapy in HIV-infected patients affects IL-7 serum levels, expression of the IL-7 receptor (CD127), and T-cell cycling. Immunophenotypic analysis of T cells from 29 HIV-uninfected controls and 43 untreated HIV-infected patients (30 of whom were followed longitudinally for ≤24 months on cART) was performed. Restoration of both CD4^+^ and CD8^+^ T cells was driven by increases in CD127^+^ naive and central memory T cells. CD4^+^ T-cell subsets were not fully restored after 2 years of cART, whereas serum IL-7 levels normalized by 1 year of ART. Mathematical modeling indicated that changes in serum IL-7 levels could be accounted for by changes in the receptor concentration. These data suggest that T-cell restoration after cART in HIV infection is driven predominantly by CD127^+^ cells and that decreases in serum IL-7 can be largely explained by improved CD127-mediated clearance.

Benito et al. [23] found that pre-cART IL-7 levels were high and interleukin-7 receptor-α (IL7Rα) expression was reduced in HIV-infected patients. This downregulation mechanism of the receptor IL7Rα is mainly related to the activation of T cells. Observations suggest that the establishment of an IL-7/IL-7R balance during HIV cART treatment can remedy the viral damage.

Ikomey et al. [24] observed higher levels of pro-inflammatory IL-2 and IL-7 in a failed cART regimen compared to lower levels of pro-inflammatory cytokines IL-2 and IL-7 in a successful cART regimen. This also suggests a possible correlation between low viral load and low pro-inflammatory cytokines. Thus, the effective decrease in viral load may have an association with decreased levels of pro-inflammatory cytokines.

Administration of IL-7 to HIV-infected patients treated with antiretroviral therapy causes a selective increase in the fraction of virgin T cells and CD4^+^ T cells, suggesting a beneficial effect on the overall function of CD4^+^ T cells [25].

In our own studies, although no statistical significance was found between IL-15 levels in HIV-infected men before and after cART and in relation to the control group, some changes in its expression were noticed. The mean concentration of IL-15 in HIV-infected patients not treated with cART was lower compared to its concentration in HIV-infected men after one year of treatment and was similar to that obtained in healthy men. 

Such a tendency was noticed by other authors studying this interleukin in the course of HIV infection Ahmad et al. [26] showed that the level of IL-15 is significantly reduced in the plasma of HIV-infected patients compared to the control group. 

Studies have shown that IL-15 is produced during acute HIV and SIV (Simian Immunodeficiency Virus) infection and can affect viremia and viral threshold. Although the role of intrinsic IL-15 during chronic infection is less defined, scientists have demonstrated in vivo that administration of IL-15 does not increase viral replication in SIV-infected animals [27].

D’Ettorre et al. [28], investigating immunological and virological interactions between IL-15 and HIV in untreated and cART-treated patients, found that the production of IL-15 by peripheral blood mononuclear cells was significantly reduced in untreated patients and patients after failed antiretroviral therapy. On the other hand, in patients who responded to cART, the production of IL-15 was comparable to that of healthy subjects. In addition, they showed that IL-15 was able to stimulate HIV-positive monocytes to produce chemokines such as IL-8 and MCP-1 (Monocyte Chemoattractant Protein-1), which specifically attract neutrophils and monocytes to the site of inflammation. This possibly improves the immune response to pathogens during HIV infection.

Albino et al. [29] unveiled that IL-15 (also IL-6 and -12) levels corresponded with immune activation after prolonged cART, thus implying that prolonged cART results in immune activation.

Structured treatment interruption (STI) may help to alleviate the problems associated with long-term cART in HIV-infected patients. A study by Amicosante et al. analyzed the role that baseline levels of cytokines in plasma play as markers of a favorable outcome of STI. Two groups of patients were defined: STI responders and STI nonresponders. STI responders showed a higher baseline concentration of IL-15 in plasma than did STI nonresponders and showed lower levels of tumor necrosis factor (TNF)-alpha during STI. No differences were observed in levels of IL-2, IL-7, or interferon-alpha in plasma. Authors showed that levels of TNF-alpha in plasma correlate with HIV viremia and monitoring baseline levels of IL-15 in plasma allows for the identification of a favorable outcome of STI [30].

By examining the interdependencies between selected interleukins, a positive correlation between IL-4 and IL-15 was demonstrated. With an increase in IL-15, an increase in IL-4 is noticeable, which may indicate that IL-15 can stimulate the production of anti-inflammatory IL-4. The obtained results may confirm the data on the role of IL-4 and IL-15 in mediating the development of specific CD8^+^ T cell memory during viral infections, which was confirmed by in vivo studies [31]. However, the exact role of this interaction in HIV infection is unknown. In our study, we showed the need for further research in this area.

The authors’ studies showed that the level of CD8^+^ T cells was higher in HIV-infected men before treatment than in patients after cART (*p* = 0.004). HIV-specific CD8^+^ T cells are prone to apoptosis, affecting their ability to control HIV infection. Since immune responses mediated by CD8^+^ T cells play a key role in controlling HIV infection, increasing the survival and effector function of HIV-specific CD8^+^ T cells may increase their ability to control HIV [14].

Our own studies did not show significant changes in the level of IL-4, -7, -15 depending on the applied cART treatment regimen (INSTIs vs. PIs). There is no information in the available literature on the study of differences in the expression of these interleukins depending on the type of antiretroviral treatment regimen.

Each cytokine plays a specific role in the regulation of a target cell, but when combined with another cytokine of the same family, this role becomes more enhanced and more effective. The ability of IL-7 and IL-15 to expand and/or enhance effector cell function may be of therapeutic benefit to HIV-infected patients. By examining the functional effects of these cytokines on HIV-specific immunity and HIV-carrier cells, researchers showed that both IL-7 and IL-15 enhance natural killer (NK) cell function more, while IL-7 enhances NK cell function by upregulating an apoptosis-inducing ligand associated with tumor necrosis factor (TNF). The close association and synergism provided by cytokines suggest the necessity to treat them as a whole during HIV treatment [32,33].

We are aware of the limitations of our work and believe that further studies should be conducted on a larger group of patients. The presented research was conducted on a group of men, and the expression of selected interleukins in women should be checked— the authors intend to analyze this matter in further research. The cART period was limited to one year; after a longer follow-up period, research should be continued to see if interleukin expression changes with long-term antiretroviral therapy.

## 5. Conclusions

The research results suggest the involvement of selected interleukins in HIV infection. The use of cART modulates the expression of the interleukins tested, especially in the case of IL-4 and IL-7. The type of treatment regimen did not significantly affect the level of interleukins in the plasma of HIV-infected patients, which may be important for the course of the disease.

Due to the existing global problem related to HIV infection, the development of AIDS, and the effectiveness of antiretroviral therapy, it seems fitting for research to explore this topic. It is for these reasons that this work was created. The obtained results, which deepen existing knowledge about changes in interleukins during the course of HIV infection, are encouraging; however, further research in this direction is needed to explain these mechanisms of action in detail.

## Figures and Tables

**Figure 1 viruses-14-00997-f001:**
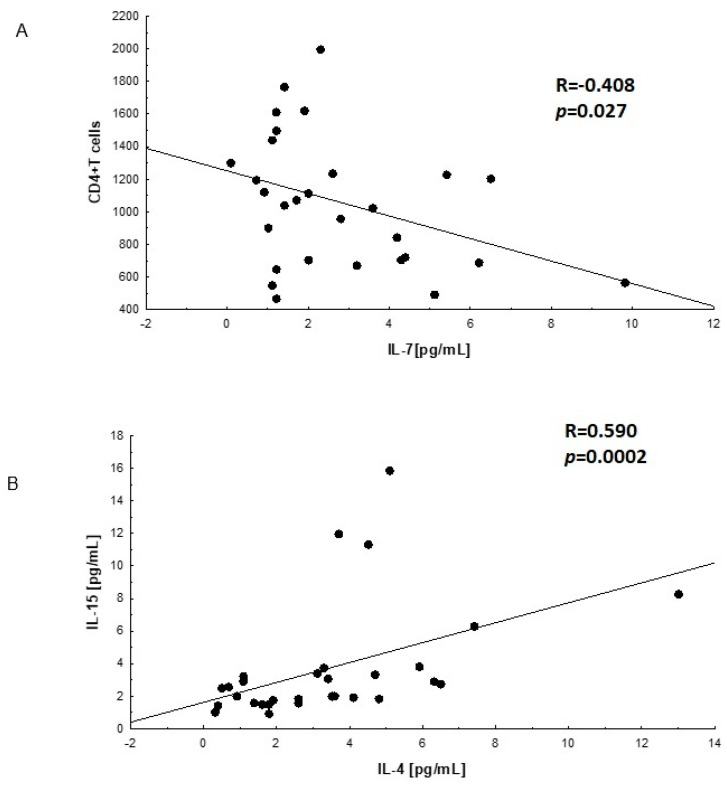
Correlations between interleukins. (**A**)—linear relationship between IL-7 concentration and the number of lymphocytes T CD4^+^ in plasma of HIV-infected men before cART; (**B**)—linear relationship between IL-4 and IL-15 in plasma of HIV-infected men after cART; the correlations were evaluated with Spearman’s non-parametric test.

**Table 1 viruses-14-00997-t001:** Analysis of immunological parameters characterizing HIV-infected men before (A) and after cART (B) depending on the HIV viral load, CD4^+^ and CD8^+^ T-cell counts.

Groups Characteristic	HIV-Infected Men before cART (A)	HIV-Infected Men after cART (B)	* *p*
	Mean (Min–Max)	Mean (Min–Max)	
HIV-RNA (copies/mL)	222 157.00 (39.00–2,030,000.00)	14.50 (0.00–85,000.00)	<0.001
CD4^+^ T cells (cells/µL)	351.00 (20.00–960.00)	600.00 (122.00–1407.00)	<0.001
CD8^+^ T cells (cells/µL)	1 094.00 (386.00–3388.00)	918.00 (257.00–2000.00)	0.004
CD4^+^/CD8^+^ ratio	0.34 (0.03–0.89)	0.75 (0.22–1.74)	<0.001

Abbreviation: N—number of men; NS—statistically insignificant value; * *p* < 0.05 statistical significance of the Wilcoxon test.

**Table 2 viruses-14-00997-t002:** Results for IL-4, -7, -15 in HIV-infected men before cART(A), after cART(B) and in the control group (C) with statistical analysis.

HIV-Infected Men before cART (A) Mean ± SD Me (IQR)	HIV-Infected Men after cART (B) Mean ± SD Me (IQR)	HIV-Uninfected Men (C) Mean ± SD Me (IQR)	*p*
IL-4 (pg/mL)			
5.39 ± 5.02 2.90 (2.00–7.00)	3.39 ± 2.68 3.20 (1.40–4.70)	2.68 ± 1.02 2.80 (2.00–3.45)	A:C = 0.03 * B:C = NS A:B = 0.04 **
IL-7 (pg/mL)			
575.00 ± 700.20 208.35 (75.00–887.40)	388.31 ± 464.27 162.40 (61.40–720.60)	383.24 ± 270.42 348.00 (133.00–508.35)	A:C = NS B:C = NS A:B = 0.02 **
IL-15 (pg/mL)			
2.56 ± 1.78 1.95 (1.20–4.20)	3.69 ± 3.58 2.55 (1.70–3.40)	2.25 ± 1.34 2.05 (1.30–3.15)	A:C = NS B:C = NS A:B = NS

Abbreviation: SD—standard deviation; IQR—(25–75%); IL—interleukin; NS—not statistically significant; The *p*-values were calculated using the Mann–Whitney *U*-test * and Wilcoxon test **; *p* < 0.05—statistically significant.

**Table 3 viruses-14-00997-t003:** Results of IL-4, -7 and -15 before (group A) and after (group B) cART in HIV-infected men by HIV-RNA viral load (below and above 100,000 copies/mL) with statistical analysis.

**HIV-Infected Men before** **cART (A)**	**HIV-RNA ≤ 100,** **000 (Copies/mL)** **(N = 16)**	**HIV-RNA > 100** **,** **000 (Copies/mL)** **(N = 14)**	** *p* **
	**Me (IQR)**	**Me (IQR)**	
IL-4 (pg/mL)	3.00 (1.85–5.10)	4.90 (2.60–7.00)	0.001 *
IL-7 (pg/mL)	224.05 (82.25–1028.20)	193.00 (74.40–1215.00)	NS
IL-15 (pg/mL)	2.15 (1.15–4.75)	1.40 (1.20–2.80)	NS
**HIV-Infected Men after** **cART (B)**	**HIV-RNA ≤ 100,000 [Copies/mL]** **(N = 30)**	**HIV-RNA > 100,000 (Copies/mL)** **(N = 0)**	** *p* **
	**Me (IQR)**	**Me (IQR)**	
IL-4 (pg/mL)	2.90 (2.00–7.00)	there were no HIV-infected men with HIV-RNA > 100,000 copies/mL in this subgroup	-
IL-7 (pg/mL)	162.40 (61.40–720.60)		
IL-15 (pg/mL)	2.55 (1.70–3.40)		

Abbreviation: Me—median; IQR—Interquartile range; N- number of participants; NS—not statistically significant; the *p*-values were calculated using the Wilcoxon test *; *p* < 0.05—statistically significant.

**Table 4 viruses-14-00997-t004:** Results of IL-4, -7 and -15 before (group A) and after (group B) cART in HIV-infected men by CD4^+^ T-cell count (below and above 300 cells/µL) with statistical analysis.

**HIV-Infected Men before** **cART (A)**	**CD4^+^ T Cells ≤ 300** **(Cells/µL)** **(N = 11)**	**CD4^+^ T Cells > 300** **(Cells/µL)** **(N = 19)**	** *p* **
	**Me (IQR)**	**Me (IQR)**	
IL-4 (pg/mL)	2.90 (1.70–7.00)	2.90 (2.60–9.10)	NS
IL-7 (pg/mL)	482.50 (89.50–1627.00)	148.30 (74.40–789.50)	0.026 *
IL-15 (pg/mL)	2.00 (1.20–4.20)	1.90 (1.10–4.30)	NS
**HIV-Infected Men after** **cART (B)**	**CD4^+^ T Cells ≤ 300** **(Cells/µL)** **(N = 4)**	**CD4^+^ T Cells > 300** **(Cells/µL)** **(N = 26)**	** *p* **
	**Me (IQR)**	**Me (IQR)**	
IL-4 (pg/mL)	4.90 (4.10–5.70)	2.60 (1.10–4.10)	NS
IL-7 (pg/mL)	118.05 (26.80–896.05)	162.40 (81.50–720.60)	NS
IL-15 (pg/mL)	3.10 (2.45–9.60)	2.25 (1.60–3.40)	NS

Abbreviation: Me—median; IQR—Interquartile range; N—number of participants; NS—not statistically significant; the *p*-values were calculated using the Wilcoxon test *; *p* < 0.05—statistically significant.

**Table 5 viruses-14-00997-t005:** Results of IL-4, -7 and -15 before (group A) and after (group B) cART in HIV-infected men by CD8+ T-cell count (below and above 1000 cells/µL) with statistical analysis.

**HIV-Infected Men before cART (A)**	**CD8^+^ T Cells ≤ 1000** **(Cells/µL)** **(N = 13)**	**CD8^+^ T Cells > 1000** **(Cells/µL)** **(N = 17)**	** *p* **
	**Me (IQR)**	**Me (IQR)**	
IL-4 (pg/mL)	3.10 (2.00–7.00)	2.90 (2.60–6.50)	NS
IL-7 (pg/mL)	789.50 (97.10–1554.00)	133.90 (74.40–269.10)	0.027 *
IL-15 (pg/mL)	3.20 (1.20–4.40)	1.40 (1.20–2.30)	NS
**HIV-Infected Men after cART (B)**	**CD8^+^ T Cells ≤ 1000** **(Cells/µL)** **(N = 9)**	**CD8^+^ T Cells > 1000** **(Cells/µL)** **(N = 21)**	** *p* **
	**Me (IQR)**	**Me (IQR)**	
IL-4 (pg/mL)	1.80 (0.50–4.10)	3.40 (1.80–4.80)	NS
IL-7 (pg/mL)	199.10 (51.20–552.80)	132.20 (87.20–720.60)	NS
IL-15 (pg/mL)	1.90 (1.50–3.30)	2.70 (1.80–3.40)	NS

Abbreviation: Me—median; IQR—Interquartile range; N—number of participants; NS—not statistically significant; the *p*-values were calculated using the Wilcoxon test *; *p* < 0.05—statistically significant.

**Table 6 viruses-14-00997-t006:** Results of Il-4, -7 and -15 in HIV-infected men after cART in the subgroup with Protease inhibitors (PIs) treatment and the subgroup with Integrase transfer inhibitors (INSTIs) treatment with statistical analysis.

HIV-Infected Men with cART			
ILs	INSTIs (N = 16) Me (IQR)	PIs (N = 14) Me (IQR)	*p*
**IL-4 (pg/mL)**	3.65 (1.00–5.35)	2.85 (1.80–3.50)	NS
**IL-7 (pg/mL)**	196.20 (66.35–636.70)	160.35 (61.40–770.00)	NS
**IL-15 (pg/mL)**	2.25 (1.70–5.05)	2.75 (1.70–3.30)	NS

Abbreviation: INSTIs—Integrase transfer inhibitors; PIs—Protease inhibitors; Me—median; IQR—Interquartile range.

## Data Availability

Not applicable.

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
