# Peer review of "Differences in Expression of Selected Interleukins in HIV-Infected Subjects Undergoing Antiretroviral Therapy"

_viruses, 2022, doi:10.3390/v14050997_

Round 1
Reviewer 1 Report
Dear Author,
thank you for the opportunity to review the manuscript entitled :"
Differences in expression of selected interleukins in HIV-in- infected subject undergoing antiretroviral therapy " by Beata Szymańska et al.
The Author explores the levels of interleukins involved in the immunity response in 30 HIV infected men before and after 1 year of cART and compare the results with a group of healthy donors.
I suggest before the publication to improve the results reported:
The paper is interesting and to improve the manuscript I suggest to Author to answer to some questions:
The mean of HIV-RNA after 1 year of cART is 20000,00;
How may patients had a failure to the therapy?
Can the Author distinguish the studied population in two group: undetectable and detectable level of HIV-RNA after 1 year of therapy?
Can the Author verify if the variation of the interleukins depends on the levels of HIV-RNA?
In the similar way the Author should analyze the difference amount of interleukins considering the increase of CD4 in 1 year; in fact the range reported is large (20.00 – 960.00) and probably the patients with lower number of CD4 has a minor modification of the levels of interleukins .
Could be interested to report the ratio CD4/CD8 and verify if the increase of the ratio is correlated to the modification of the levels of inteleukins.
In fact the range of CD8 is large.
Author Response
Answers to comments to manuscript viruses-161914 Reviewer 1.
Thank you very much for sending referee’s comments on our manuscript entitled: Differences in expression of selected interleukins in HIV-in- infected subject undergoing antiretroviral therapy.
Please find enclosed our revised manuscript. We have revised the manuscript accordingly and provide specific answers below. We responded to all comment. New changes to the manuscript have been marked up using the “Track Changes”. We tried to follow all suggestions and we hope that they have allowed us to improve our manuscript. I am writing also on behalf of the remaining authors, who support submission of the revised form of the manuscript.
Dear Author,
thank you for the opportunity to review the manuscript entitled :"Differences in expression of selected interleukins in HIV-in- infected subject undergoing antiretroviral therapy " by Beata Szymańska et al.The Author explores the levels of interleukins involved in the immunity response in 30 HIV infected men before and after 1 year of cART and compare the results with a group of healthy donors.I suggest before the publication to improve the results reported.The paper is interesting and to improve the manuscript I suggest to Author to answer to some questions:
The mean of HIV-RNA after 1 year of cART is 20000,00;
Answer: Thank you for your attention, text in the table has been corrected: was 20000 is 2000. (Table 1, page 4).
How may patients had a failure to the therapy?
Answer: After one year of cART therapy, the level of HIV RNA in 11 HIV infected men decreased (mean number 40 copies / mL). In 19 HIV-infected men HIV RNA were undetectable. To sum up, in 63% of HIV-infected men, the HIV virus was undetectable, which proves the high effectiveness of cART.
Can the Author distinguish the studied population in two group: undetectable and detectable level of HIV-RNA after 1 year of therapy?
Answer: HIV infected men after one year of cART therapy had less than 100 000 copies/ mL of HIV RNA. In 19 HIV infected men HIV RNA level was undetectable.
Can the Author verify if the variation of the interleukins depends on the levels of HIV-RNA?
Answer: Changes in ILs depending on the level of HIVRNA below and above 100 000 copies/mL both, before and after cART are presented in Table 3. Additionally, following the reviewer's suggestion, interleukin levels were compared in HIV-infected men after treatment (B) with undetectable HIV RNA (n = 19) to HIV infected men after cART with detected viremia (n = 11; mean 40 HIV RNA copies /mL). New information has been added in subchapter 3.4 page 7, lines 244-253.
In the similar way the Author should analyze the difference amount of interleukins considering the increase of CD4 in 1 year; in fact the range reported is large (20.00 – 960.00) and probably the patients with lower number of CD4 has a minor modification of the levels of interleukins.
Answer: The analysis is presented in Table 4. Precisely for this reason, after 1-year cART HIV-men (B) was divided into two groups: below and above 300 cells / µL CD4 + and the results are summarized in Table 4.
Could be interested to report the ratio CD4/CD8 and verify if the increase of the ratio is correlated to the modification of the levels of interleukins.
Answer: As suggested by the Reviewer, the information on the value of interleukins depending on the CD4 + / CD8 + ratio was supplemented. Subchapter 3.7 has been added. Page 9, lines 309-328.
In fact the range of CD8 is large.
Answer: We agree with the Reviewer, but we took this into account when analyzing HIV-men scores (both in groups A and B) in two subgroups ≤and> 1000 cells /ul.
Thank you very much for these valuable comments and we hope that the amendments will significantly improve the quality of our manuscript and enable its publication in Viruses.
Thank you very much for your time.
Reviewer 2 Report
This article concerns the study of some plasmatic cytokines in HIV infected individuals before or after starting anti-retroviral treatment. The authors concluded that ARV treatment modifies the levels of IL-4 and IL-7 in this group of infected individuals. Although the results presented here in have some interest, several critical points need clarifications by the authors.
Comments:
- One important point is what is new. As several articles concerning the plasmatic cytokines and their modification with or without medical intervention have been already published, the authors must point out the “new” results presented here.
- The choice of tested cytokines is not well explained.
- The heterogenicity of the individuals makes difficult the interpretation. This will be taken in account in the discussion.
- Materials and Methods will be Individuals and Methods.
- Lines 95 – 98: One of the inclusion criteria for HIV infected individuals is “taking cART drugs”. If I understood well the plasmatic levels of cytokines were measured before and after treatment. Exclusion criteria will be non-inclusion criteria.
- Lines 100 – 102: if the volunteers are healthy that means they do not have any diseases. This will be modified.
- Lines 106 – 109: analysis before or after treatment?
- Lines 110 – 111: “data on the number” will be “data on the levels”. A short description of the method for T cells measurement will be added as well as for HIV viral load with the limit of the detection.
- Lines 142 – 146: HIV viral load is not an “immunological parameter”. Thus, virological and immunological parameters will be used.
- In table 1 is indicated “20000 CD8 cells”, is it correct?
- There is a strong overlapping in the values shown in the tables 1 to 4. The tables must be replaced by the figures in which individual points are shown to have a better idea of the differences between the groups.
- In figure 1, the R values are very low and limit the extend of this observation (< 0.35 are generally considered to represent low or weak correlations, 0.36 to 0.67 modest correlations, and 0.68 to 0.89 strong correlations and > 0.90 very high correlation). P value is missing in figure 1B. And in line 291 is indicated p=0.000 (???)
- It is not clear what additional information the figure 2 gives.
- The signification of the correlation between IL-4 and IL-15 plasmatic levels is not evident and have to be discussed better.
- As multiple comparisons were done, a statistical corrector test will be applied (ex. Bonferroni test or other)
- In the discussion, a clear difference must be done between cytokine plasmatic levels in acute/early versus chronic HIV infection. Still here, as the immunological parameters are very large, it is not easy to classify HIV infection.
- Line 401: it is not clear the relationship between a decrease in the number of CD8 T cells and the correlation observed between IL-4 and IL-15.
- In my opinion, the best way to correlate immunological parameters is to calculate the gain (for CD4+ cells) or loss (for CD8+ cells) per day and correlate them individually with the cytokine values after treatment. As the results are presented here, only HIV viral load seems to have an effect in the levels of cytokines measured in the plasma.
Author Response
Thank you very much for sending referee’s comments on our manuscript entitled: Differences in expression of selected interleukins in HIV-in- infected subject undergoing antiretroviral therapy.
Please find enclosed our revised manuscript. We have revised the manuscript accordingly and provide specific answers below. We responded to all comment. New changes to the manuscript have been marked up using the “Track Changes”. We tried to follow all suggestions and we hope that they have allowed us to improve our manuscript. I am writing also on behalf of the remaining authors, who support submission of the revised form of the manuscript.
This article concerns the study of some plasmatic cytokines in HIV infected individuals before or after starting anti-retroviral treatment. The authors concluded that ARV treatment modifies the levels of IL-4 and IL-7 in this group of infected individuals. Although the results presented here in have some interest, several critical points need clarifications by the authors.
Comments:
One important point is what is new. As several articles concerning the plasmatic cytokines and their modification with or without medical intervention have been already published, the authors must point out the “new” results presented here.
- The choice of tested cytokines is not well explained.
Answer: As suggested by the Reviewer, information on ILs in relation to HIV was added in the introduction.
Page 2, lines 67-77, lines 82-89, lines 97-108
- The heterogenicity of the individuals makes difficult the interpretation. This will be taken in account in the discussion.
Answer: The study was conducted on a group of HIV infected men (30) and healthy men (28) of similar size and age, so we do not understand the allegation of heterogeneity of individuals. In further research, we want to increase the study group and broaden it to include women, and to analyze the results after a cART period longer than a year.
- Materials and Methods will be Individuals and Methods.
Answer: As suggested, the chapter title has been changed
- Lines 95 – 98: One of the inclusion criteria for HIV infected individuals is “taking cART drugs”. If I understood well the plasmatic levels of cytokines were measured before and after treatment. Exclusion criteria will be non-inclusion criteria.
Answer: We agree with the Reviewer, the information has been corrected.
- Lines 100 – 102: if the volunteers are healthy that means they do not have any diseases. This will be modified.
Answer: As suggested, the sentence was corrected.
- Lines 106 – 109: analysis before or after treatment?
Answer: The information was supplemented with:
“Patients were also analyzed in subgroups depending on HIV RNA viral load (below and above 100,000 RNA copies / mL), CD4 + T cell count (below and above 300 cells / µL), CD8 + T cell count (below and above 1000 cells / µL) and the type of therapeutic regimen (INSTIs or PIs) both before and after one year of anti-HIV therapy.”
- Lines 110 – 111: “data on the number” will be “data on the levels”. A short description of the method for T cells measurement will be added as well as for HIV viral load with the limit of the detection.
Answer: Information on the LT measurement method and the determination of the level of HIV RNA with the limit of detection has been added:
“HIV-RNA isolation was performed using a system viral nucleic acid kit (Roche Diagnostics, Mannheim, Germany). HIV-1 viral load was measured by real-time PCR assay (COBAS TaqMan HIV-1 test v. 2.0, Roche Diagnostics, Basel, Switzerland). Detection limit was 40 copies/ml. The CD4+ and CD8+ T cell count were tested by flow cytometry with the use FacsCount Becton Dickinson system (BD Biosciences, San Jose, California, United States).”
Lines 142 – 146: HIV viral load is not an “immunological parameter”. Thus, virological and immunological parameters will be used
Answer: As suggested, we removed the incorrect wording.
- In table 1 is indicated “20000 CD8 cells”, is it correct?
Answer: Obviously the value was 2000 CD8 + cells, the mistake was corrected
- There is a strong overlapping in the values shown in the tables 1 to 4. The tables must be replaced by the figures in which individual points are shown to have a better idea of the differences between the groups.
Answer: Thank you for suggesting a different interpretation of the results, but we will stick to the original version, counting on the Reviewer's favor. We want to keep the tables, because we can show the exact values of the median concentration and IQR of the parameters tested, which allows for an accurate interpretation of the results. We also added graphs in Supplementary materials.
- In figure 1, the R values are very low and limit the extend of this observation (< 0.35 are generally considered to represent low or weak correlations, 0.36 to 0.67 modest correlations, and 0.68 to 0.89 strong correlations and > 0.90 very high correlation). P value is missing in figure 1B. And in line 291 is indicated p=0.000 (???)
Answer: We realize that the correlation is average, but the relationship is statistically significant, so we decided to present this results in the article.
Interpretation of the Spearman correlation coefficient (R):
average correlation (significant correlation) 0.3 ≤ r <0.5
high correlation (significant correlation) 0.5 ≤ r <0.7
very high correlation (very high dependence) 0.7 ≤ r <0.9
full correlation 0.9 ≤ r <1.0
- It is not clear what additional information the figure 2 gives.
Answer: Additional interpretation of the results in a graphic form makes it easier and faster to observe the differences in the results between groups. We placed the graphs as supplementary material.
- The signification of the correlation between IL-4 and IL-15 plasmatic levels is not evident and have to be discussed better.
Answer: Our study showed a positive correlation between the levels of IL-4 and IL-15. The precise significance of the interaction of these two interleukins in the course of HIV infection is unknown. Our study shows the need for further research in this area.
We added short information to the discussion section: „The obtained results may confirm the data on the role of IL-4 and IL-15 in mediating the development of specific CD8 + T cell memory during viral infections, which was confirmed by in vivo studies[32]. However, the exact role of this interaction in HIV infection is unknown. In our study, we showed the need for further research in this area.”
- As multiple comparisons were done, a statistical corrector test will be applied (ex. Bonferroni test or other)
Answer: The Wilcoxon test was used to compare dependent groups, which was shown in the chapter "Statistical analysis”.
- In the discussion, a clear difference must be done between cytokine plasmatic levels in acute/early versus chronic HIV infection. Still here, as the immunological parameters are very large, it is not easy to classify HIV infection.
Answer: We agree with the reviewer. In our study, we have divided HIV infected men into subgroups according to lower and higher CD4 +, CD8 + cell counts, and HIV RNA levels in order to better interpret the obtained interleukin values. This is a preliminary study, more people are needed to interpret clear differences. In our opinion, the division used in terms of immunological parameters and viremia is the most reliable. We do not have data that would confirm that the patient was in the acute phase of infection at the time of diagnosis.
Line 401: it is not clear the relationship between a decrease in the number of CD8 T cells and the correlation observed between IL-4 and IL-15.
Answer: Information on the number of CD8 + is not associated with the presence of a correlation between IL-4 and IL-15. As suggested, we corrected the text by placing this data in separate paragraphs.
- In my opinion, the best way to correlate immunological parameters is to calculate the gain (for CD4+ cells) or loss (for CD8+ cells) per day and correlate them individually with the cytokine values after treatment. As the results are presented here, only HIV viral load seems to have an effect in the levels of cytokines measured in the plasma.
Answer: We agree with the reviewer's suggestion that an analysis comparing patients with or without improvement in CD4 + and CD8 + T cell count would be useful, but it is not possible to perform this type of analysis, because most of the patients had acceptable levels of CD4 +, CD8 + cells after treatment. The authors created subgroups of patients before and after cART, based on CD4 + and CD8 + T cell count. The authors compared levels of IL-4, IL-7 and IL-15 in patients with high CD4 + or CD8 + T cell count in the group before and after cART, and the same comparison was made in patients with low CD4 + and CD8 + T cell count, as suggested by the attending physician, monitoring the patients. In our opinion, it will also be reasonable to conduct further analyses and increase the size of the studied population, which is planned in our further research. We also plan to include patients with therapy failure and no improvement in immune parameters or an altered therapeutic regimen in case of therapy failure.
In the next research, we will use a very interesting way of interpreting the results proposed by the reviewer.
Thank you very much for these valuable comments and we hope that the amendments will significantly improve the quality of our manuscript and enable its publication in Viruses.
Thank you very much for your time.
Round 2
Reviewer 2 Report
Dears
I do not have more comments